# Abstraction and Idealization in Biomedicine: The Nonautonomous Theory of Acute Cell Injury

**DOI:** 10.3390/brainsci8030039

**Published:** 2018-02-27

**Authors:** Donald J. DeGracia, Doaa Taha, Fika Tri Anggraini, Shreya Sutariya, Gabriel Rababeh, Zhi-Feng Huang

**Affiliations:** 1Department of Physiology, Wayne State University, Detroit, MI 48201, USA; dr.fikatri@yahoo.co.id; 2Department of Physics and Astronomy, Wayne State University, Detroit, MI 48201, USA; dotaha@umich.edu (D.T); fp4165@wayne.edu (S.S); fy1826@wayne.edu (G.R); huang@wayne.edu (Z-F.H)

**Keywords:** acute cell injury, brain ischemia, cell death, nonautonomous differential equation, neuroprotection, preconditioning

## Abstract

Neuroprotection seeks to halt cell death after brain ischemia and has been shown to be possible in laboratory studies. However, neuroprotection has not been successfully translated into clinical practice, despite voluminous research and controlled clinical trials. We suggested these failures may be due, at least in part, to the lack of a general theory of cell injury to guide research into specific injuries. The nonlinear dynamical theory of acute cell injury was introduced to ameliorate this situation. Here we present a revised nonautonomous nonlinear theory of acute cell injury and show how to interpret its solutions in terms of acute biomedical injuries. The theory solutions demonstrate the complexity of possible outcomes following an idealized acute injury and indicate that a “one size fits all” therapy is unlikely to be successful. This conclusion is offset by the fact that the theory can (1) determine if a cell has the possibility to survive given a specific acute injury, and (2) calculate the degree of therapy needed to cause survival. To appreciate these conclusions, it is necessary to idealize and abstract complex physical systems to identify the fundamental mechanism governing the injury dynamics. The path of abstraction and idealization in biomedical research opens the possibility for medical treatments that may achieve engineering levels of precision.

## 1. Introduction

Many important clinical conditions continue to elude effective treatments. Stroke is a notable example where over 100 clinical trials have failed to find a means to prevent cell death by neuroprotection [1]. Similar strings of clinical trial failure have occurred with diseases such as myocardial infarction [2] and acute nephrotic ischemia [3]. We have argued that a key factor behind these failures is the lack of a general theory of biological cell injury.

We therefore introduced a nonlinear dynamical theory of acute cell injury [4]. However, the original form of the theory possessed limitations as detailed in Ref. [5]. This led us to reformulate the theory, which was introduced elsewhere [5] but for which we here provide a deeper analysis. Technically, the original theory consisted of a system of *autonomous* nonlinear ordinary differential equations. The reformulation of the theory consists of a system of *nonautonomous*, nonlinear differential equations. The technical mathematical differences impact how the equations are solved and how the resulting solutions are interpreted in terms of acute cell injury. It is the purpose of this paper to illustrate our procedure for solving and interpreting the solutions of the nonautonomous theory of acute cell injury. 

As the title indicates, constructing and understanding the theory, whatever mathematical form it takes, requires abstracting and idealizing the real world. Biology and the subdisciplines of biomedicine are generally descriptive. Advanced mathematics are not widely used in these sciences. On the other hand, the sciences that have allowed the most effective development of technology, notably physics, do not seek to literally describe its subject matter. Instead, physics idealizes and abstracts reality to mathematically formulate how things change. The second major aspect of the present work is to illustrate how idealization and abstraction can be used in biomedical research to model acute injury to biological cells.

In this regard, a key idea we wish to convey is that *mathematically abstracting a system allows all possible states of the system to be understood*. Experiments usually involve high costs of time and resources and can only measure a finite number of parameter combinations. On the other hand, a theory allows us to see how the system behaves under all parameter conditions. Then, the goal of science is to assure congruency between specific solutions of the theory and specific experimental measurements. Experimental measurements made only to describe a phenomenon in the absence of mathematical theory are incomplete and are descriptive science [6]. Rather, the measurements should seek to test a theory [6].

Below we study one idealized cell type injured by one idealized injury mechanism. We shall see the enormous complexity in this one example. However, the complexity is not incomprehensible. The mathematics provide a systematic framework, a catalog of sorts, allowing all possible states of the system to be understood in an organized fashion. This has major implications for developing therapies for acute injuries such as stroke. We shall show that injuring one ideal cell type by one ideal injury mechanism, varying only the intensity of the injury, produces a continuum of states for which there is little appreciation in the paradigms that currently dominate biomedical research.

### 1.1 Cell Injury Idealized 

There are two main categories of how cells become injured. There is either some identifiable injury applied to the cells or there is not. Examples of identifiable injuries would include mechanical trauma, chemical trauma (e.g., a poison), metabolic trauma (such as ischemia), and so on. The criteria being that there is a clearly identifiable exogenous agent or circumstance that injures the cell. Further, the intensity of the damage mechanism can be quantified: the minutes of ischemia, the concentration of a poison, the amount of force, and so on. We can abstract the quantitative aspect of *injury intensity* as a parameter *I*, and thereby abstract it from any specific injury mechanism. This stands in contrast to injuries such as cell transformation, or chronic neurodegenerative diseases in which the cause, let alone intensity, of the cell injury is not clearly identifiable, if even known. Thus, *I* represents the intensity of a clearly identifiable injury mechanism and we term this “acute cell injury”.

From these considerations, we can begin to construct an idealized picture of acute cell injury. A cell is injured by some acute injury mechanism, and in response it either lives or dies. This is an idealization because we do not specify what type of cell nor the specific injury. The cell and the injury are tokens that interact, and their interaction is quantified by the parameter *I*.

From the vast amount of empirical, descriptive studies, it has been shown that many biomolecular changes occur in cells after they have been acutely injured. These are generally expressed in terms of pathways: changes in phosphorylation or other signaling events, increases or decreases in the activity of specific proteins or pathways, changes in localization or amounts of ions, specific proteins, transcription factors, micro-RNAs, etc. It is these specific molecular events that constitute the complex network of changes in the injured intracellular milieu. It is a case of Humpty Dumpty: we injure cells (or tissues) then grind them up and identify the hundreds of changes in the biomolecules, but how to put them back together again to reconstruct how the cell dies? To date, such reconstruction efforts have generally been unsuccessful as attested by failed clinical trials in many fields of biomedicine.

Instead of a literal reconstruction of the events, we idealize as follows. We know *a priori* that some of these biomolecular changes harm the cell but that others serve to protect the cell. Let all the cell-damaging changes be represented by the variable *D*, the total damage in the cell. Likewise, represent all the pro-survival changes in the cell by the variable *S*, the total stress response. Theoretically, there is a third category of changes in the acutely injured cell: those changes that have no effect on damaging the cell or helping it survive, which we denote by an empty set [ ]. Thus, all the complex molecular changes in the cell fall into these three general categories: *D*, *S*, or [ ]. There are no other possibilities. Since [ ] has no effect on outcome, we can ignore it here.

The concepts of *D* and *S* are key abstractions and idealizations of acute cell injury. We assert they provide an incontrovertible generalization that covers all possibilities. In mainstream biomedical research, the goal is to discover *which* molecules or pathways damage, and *which* enhance cell survival. Further, do some molecules that, at one point in the post-injury time course foster survival, transform into damaging influences? We submit that such questions lead down blind alleys. As abstractions, *D* and *S* already implicitly subsume these possibilities. From a theoretical perspective, we do not need to know the specific molecules of which *D* and *S* consist at any moment in time any more than we need to know the exact velocity of each air molecule when we measure air temperature.

Thus, we can fill in our ideal picture further: There is a cell. It is acutely injured with intensity *I*. In response, inside the cell, *D* and *S* assume nonzero values that change with time. The changes in *D* and *S* directly determine the survival or death outcome.

### 1.2 What Causes Cell Death? 

The main use to which the concepts of *D* and *S* are applied is *to define the cause of cell death*. The core assumption of the mathematical idealization of acute cell injury is this: if *S* > *D*, the cell lives but if *D* > *S*, the cell dies.

A metaphor may help illustrate the concept. Imagine kicking a ball up a hill. The ball starts at a position at the bottom of the hill. This is the uninjured cell. A force is applied to the ball and it rolls up the hill. The force is the application of injury *I* to the cell. If the force is weak, the ball will roll to some point on the facing hillside, then roll back down to where it started. This is the survival case where *S* > *D*. However, if the force of *I* is greater than some specific value, the ball will go over the top of the hill and roll down the other side. This is the state *D* > *S*, and the other side of the hill is the state of death.

To be more precise, we are suggesting that acute cell injury is a tipping point phenomenon. This is already recognized in other terms by the concept of “cell death threshold”. The “cell death threshold” is the amount of injury that causes the cell to die. The “cell death threshold” concept does not describe a threshold but instead indicates a *tipping point* between survival and death. Thresholds and tipping points are different mathematical entities, as illustrated below. The tipping point between survival and death is quantitatively determined by the intensity of injury *I*.

Another useful way to give meaning to *D* and *S* is to recognize that, before injury, the cell is in a homeostatic steady-state. Application of injury intensity, *I*, “knocks” the cell out of homeostasis. The variable *S* represents changes inside the cell seeking to bring the cell back into homeostasis. *D* represents changes that disrupt homeostasis. If the disruption of homeostasis is greater than the cell’s ability to re-achieve homeostasis (e.g., *D* > *S*), the cell dies.

### 1.3 Survival and Death Outcomes after Acute Injury

These ideas intimately link sublethal and lethal injuries which, however, are generally studied separately in mainstream biomedical studies, and generally treated as different phenomena. On the sublethal side is survival after injury, often accompanied by a preconditioning effect [7,8,9,10,11]. Preconditioning is specifically defined here to mean that a cell subjected to a sublethal injury can, after a specified time, survive a lethal injury. On the lethal side is cell death which can take on different qualitative forms.

With respect to cell death after brain ischemia, necrosis and delayed neuronal death (DND) [12,13] are observed. They appear different by many criteria and therefore are thought to be due to different causes. Thus, one finds terms such as apoptosis, necrosis, and necroptosis, and other variants, described by lists of qualitative features such as cell appearance during death or which molecular pathways are activated [14,15,16,17,18,19,20,21,22,23].

With respect to survival and preconditioning, empirical evidence shows that the time between the sublethal and lethal injuries required to achieve optimal survival is finely tuned [24,25,26]. Thus, there are degrees of preconditioning, and the preconditioning effect is not permanent but fades with time after the sublethal injury. Further, different tissues display different forms of preconditioning (e.g., rapid preconditioning in heart, and delayed preconditioning in brain [27]).

Rapid or delayed preconditioning, necrosis and DND are qualitative distinctions. By defining acute cell injury in terms of the dynamics of *D* and *S* we build a framework where these seemingly different phenomena are points on a continuum of responses of cells to acute injury. The continuum is injury intensity, *I*. The dynamics of *D* and *S* vary quantitatively as a function of *I* as we will show below. This allows us to understand clearly that the specific qualitative forms of preconditioning or cell death are but cross sections of a continuum of dynamical behaviors.

Thus, through idealization and abstraction, we define the cause of cell death in a general fashion, independent of any specific cell type or specific injury mechanism, by focusing on and abstracting features common to all cells. By converting these ideas into a mathematical theory, we intimately link survival and death outcomes in a single continuum of dynamical behaviors.

### 1.4. A Mathematical Theory of Acute Cell Injury

We have developed the refined theory elsewhere [5] and therefore give only a brief summary here. We begin with our idealized picture of acute cell injury: there is a cell which is acutely injured with intensity *I*. *D* and *S* accumulate in the cell and change with time. If *D* > *S*, the cell irreversibly exits homeostasis and dies. 

By definition, *D* and *S* are mutually antagonistic. Stress responses seek to eliminate damage, but damage reactions can destroy the mediators of stress responses. This has a specific mathematical meaning: *D* and *S* are inversely related. The inverse relationships between *D* and *S* can be quantified using the Hill function, which is the function that gives S-shaped curves. The Hill function defines a threshold, *Θ*, where a 50% effect occurs (e.g. like LD_50_). Thus, there is some amount of *S*, *Θ_S_*, which causes a 50% decline in *D*. Similarly, there is some amount of *D*, *Θ_D_*, which causes a 50% reduction in *S*. *Θ_D_* and *Θ_S_* are true mathematical thresholds, meaning they are the values of *D* and *S* at the 50% reduction point.

The key assumption of our theory is that *Θ_D_* and *Θ_S_* change as a function of injury intensity, *I*.
(1)$$\begin{array}{l} \Theta_{D}^{}=c_{D}Ie^{I\lambda_{D}}\\ \Theta_{S}^{}=c_{S}Ie^{-I\lambda_{S}} \end{array}$$

Equation (1) posits specific functional forms: *Θ_D_* increases and *Θ_S_* decreases exponentially with *I*. These constitute assumptions that require empirical verification. However, for the sake of theory building, they provide a simple, plausible relationship. The parameters *c_D_*, *c_S_*, *λ_D_*, and *λ_S_* are constants of proportionality, required to correctly express the proportionalities between the thresholds and *I*. Their meanings are discussed in the next section.

To model the changes in *D* and *S* with time, the following equation is used
(2)$$\begin{array}{l} \frac{dD}{dt}=v_{D}\frac{\Theta_{D}^{n}}{\Theta_{D}^{n}+S^{n}}-k_{D}D\\ \frac{dS}{dt}=v_{S}\frac{\Theta_{S}^{n}}{\Theta_{S}^{n}+D^{n}}-k_{S}S\\ net=formation-decay\\ rate\;\;rate\;\;rate \end{array}$$

As indicated, this is a well-known system of differential equations for modeling the common sense understanding that the net rate equals the difference between the rate of formation and the rate of decay [28]. The rates of formation are given by Hill functions expressing the inverse relationship between *D* and *S*, scaled by a velocity parameter *v*. In Equation (2), the rates of decay are assumed to be linear, with a decay parameter *k*.

The original theory substituted Equation (1) into Equation (2) [4]. We present here a refined theory using two additional assumptions. In addition to the assumption embodied by Equation (1) we now assume that (a) the velocity parameter decreases exponentially with time (Equation (3)), and (b) the decay parameter is a function of the instantaneous difference of *D* and *S* (Equation (4)). The rationalization of these assumptions was discussed previously [5].
(3)$$v_{D}=v_{S}=v_{0}e^{-c_{1}t}$$
(4)$$k_{D}=k_{S}=c_{2}|D-S|$$

Substituting Equations (1), (3) and (4) into Equation (2) gives our current theory of the nonlinear dynamics of acute cell injury:(5)$$\begin{array}{c} \frac{dD}{dt}=v_{0}e^{-c_{1}t}\frac{{\left(c_{D}Ie^{I\lambda_{D}}\right)}^{n}}{{\left(c_{D}Ie^{I\lambda_{D}}\right)}^{n}+S^{n}}-c_{2}|D-S|\cdot D\\ \frac{dS}{dt}=v_{0}e^{-c_{1}t}\frac{{\left(c_{S}Ie^{-I\lambda_{S}}\right)}^{n}}{{\left(c_{S}Ie^{-I\lambda_{S}}\right)}^{n}+D^{n}}-c_{2}|D-S|\cdot S \end{array}$$

## 2. Methods

### 2.1. Preliminary Considerations 

When the term “prediction” is used in a scientific context, it does not refer to qualitative statements. The term “prediction” specifically means that one of the mathematical solutions of a mathematical theory fits a dataset intended to measure the theory. A necessary precondition to data fitting is to interpret the theory in terms of the relevant physical system. Our goal now is to display a typical solution of Equation (5) and how it can be interpreted with respect to experimental situations.

To summarize what is detailed below: Equation (5) takes input numbers (the “input vector”) and outputs time courses of *D* and *S*. A time course can begin at *D* = 0 and *S* = 0, corresponding to an uninjured system, or it can begin at any value of *D* or *S*. Where the time courses start are the *initial conditions*, (*D_0_*, *S_0_*), which are part of the input vector. For a given pair of *D* and *S* time courses output by Equation (5), either *D* or *S* achieves a maximum value (*D_max_* and *S_max_*, respectively). Above we spoke of *S* > *D* or *D* > *S*. In the time courses output by Equation (5), these inequalities take the form *S_max_* > *D_max_* or *D_max_* > *S_max_*, corresponding to survival and death outcomes, respectively. The *control parameter* is *I*, injury intensity, and all other parameters are held constants to examine how the system behaves as a function of injury intensity. The series of time course along the continuum of *I* we have termed an *injury course* [4]. We show below how to express the solutions of Equation (5) as injury courses calculated across a range of initial conditions.

### 2.2 The Input Vector 

Equation (5) has 9 parameters and two initial conditions (*D_0_*, *S_0_*), giving the following input vector:
[*v_0_*, *c_1_*, *c_2_*, *c_D_*, *λ_D_*, *c_S_*, *λ_S_*, *n*, *I*, *D_0_*, *S_0_*]

A brief description of the parameters now follows. *v_0_* is the initial velocity of *D* and *S* formation at time zero. As Equation (3) indicates, the initial velocity decreases exponentially with time, meaning that the rate at which *D* and *S* form decreases with time after the injury. *c_1_* is a decay constant indicating how quickly *v_0_* decreases with time; the larger *c_1_*, the faster *v_0_* decreases. *c_2_* sets the rate that *D* and *S* decay; the larger *c_2_*, the faster are the *D* and *S* decay rates. In general, if *v_0_* and *c_2_* are set to 1, the solutions to Equation (5) stay in the unit plane (i.e., *D* and *S* range only between 0 and 1).

Four parameters, *c_D_*, *λ_D_*, *c_S_*, and *λ_S_*, characterize the qualitative aspects of the system. As described in detail elsewhere [29], *c_D_* and *λ_D_* represent the injury mechanism, and *c_S_* and *λ_S_* represent the cell type. The parameter *n*, typically called the Hill coefficient, can, in the context of our theory, be taken to represent how “tightly” the nodes of the molecular networks defined by *D* and *S* are linked. As stated above, *I*, injury intensity, is the control parameter. All these parameters can vary over large ranges and produce sensible output from Equation (5).

### 2.3 Initial Conditions

Initial conditions (*D_0_*, *S_0_*) were described above but merit further discussion because they (1) link to experimental designs commonly encountered in biomedical studies, and (2) provide an example where a mathematical theory can study situations that are limited by time and resources in the laboratory. 

In the laboratory, there is generally an uninjured control condition that is compared against the injured cells. A typical example would be a cell culture given a poison (e.g., thapsigargin that inhibits the endoplasmic reticulum SERCA pump). In this instance, thapsigargin is the injury mechanism, and its concentration is the injury intensity, *I*, which can be sublethal or lethal. For such a study there will always be a control cell culture given only the vehicle in which the thapsigargin is dissolved. Prior to administering thapsigargin, the experimental cells are identical to the control cells. This is an example of beginning an injury from initial conditions (*D_0_*,*S_0_*) = (0,0). Prior to drug administration, there is no cell damage and no active stress responses in the cell culture. 

However, what if the cells were first transfected to express, say, HSP70 protein? HSP70 is an important pro-survival stress response protein. One would hypothesize that the transfected cells should survive a higher concentration of thapsigargin than the un-transfected cells. Increasing HSP70 protein before administering thapsigargin means that *S_0_* is no longer zero but is now a positive value greater than zero. Transfecting HSP70 therefore represents a change in the *S* initial conditions of the cells. In the typical case, the experimental group would be transfected plus thapsigargin, and the control would be transfected plus vehicle. However, the transfected control is different from the untreated, un-transfected control. It is generally assumed that it is good enough to take the transfected plus vehicle as the control, and the effect of transfecting with HSP70 is not taken into account in the study design. However, in the scope of our model, the transfection of the control cells is a change in initial conditions. As we show below, changing initial conditions can radically alter the dynamics.

In general, experimental manipulations such as the example given above are not recognized as changes in initial conditions, and hence, these manipulations are not systematically accounted for in the empirical biomedical literature. The study of varied initial conditions provides a systematic handle on such circumstances. Equation (5) can be studied over ranges of initial conditions that would otherwise require prohibitive amounts of time and resources to empirically study in the laboratory. Thus, at any value of *I*, we also study the behavior of Equation (5) over a range of initial conditions.

It is theoretically relevant to ask: what dictates the range of initial conditions? Across an injury course, there will be one time course that gives the maximum possible value for *D* and another that gives the maximum possible value of *S*. The maximum value of *S* across all time courses can be interpreted as the maximum possible total stress responses for the specific cell type. Therefore, *S* cannot exceed this value. Thus, any initial condition of *S* must be less than or equal to this maximum value. For example, if the maximum *S* across all time courses = 1, then the initial condition for *S* must range as 0 < *S_0_* < 1. The same logic holds for *D_0_*. There are other possibilities for setting the initial condition ranges, but this one will serve in the present analysis.

### 2.4. Summary of Input Vector

For the present exercise, we hold [*v_0_*, *c_1_*, *c_2_*, *c_D_*, *λ_D_*, *c_S_*, *λ_S_*, *n*] constant. We then vary *I* over the range 0 < *I* < *I_max_*, where *I_max_* is the injury intensity beyond which the cells are incapable of mustering any stress response (e.g., *S* ~ 0) for the entire post injury time course. The behavior of the system over the range 0 < *I* < *I_max_* constitutes the *injury course*. The *I*-range is determined relative to *I_X_*, the tipping point value of injury intensity [4]:(6)$$I_{X}=\frac{\ln(c_{S})-\ln(c_{D})}{\lambda_{D}+\lambda_{S}}$$

Finally, at each value of *I*, within the *I*-range, we study Equation (5) over ranges of initial conditions. To repeat, if *v_0_* and *c_2_* each equal 1, then the values of *D* and *S* over time never exceed 1, and so our initial conditions can be confined to the range 0–1. From initial conditions (0,0), all time courses from *0 < I< I_X_* will have *S_max_* > *D_max_* (survival outcome), and all time courses from *I_X_* < *I* < *I_max_* will have *D_max_ > S_max_* (death outcome). This statement, however, does not hold in general at other initial conditions. We show below how to represent outcomes across ranges of initial conditions at each value of *I*. 

To summarize, the values of parameters and initial conditions used in our example are:[*v_0_*, *c_1_*, *c_2_*, *c_D_*, *λ_D_*, *c_S_*, *λ_S_*, *n*] = [1, 1, 1, 0.1, 0.25, 0.4, 2, 4].*I*-range centered at *I_X_* = 0.6161 (as calculated from the parameters given in 1).Maximum possible range of initial conditions: 0 < *D_0_* < 1 and 0 < *S_0_* < 1.

## 3. Results

### 3.1. Solutions to the Theory of Acute Cell Injury 

Equation (5) was custom programmed into Matlab (Mathworks, Natick, MA, USA, version 9.0) and solved using the ode45 solver which implements a variant of the Runge-Kutta method. In this section, we proceed in the following stages:Display time courses (and the corresponding trajectories) at specific values of *I*.Display the injury course where each time course begins at (*D_0_*, *S_0_*) = (0, 0).Display time courses at a specific value of *I* and a range of initial conditions.Display the injury course across a range of initial conditions.

### 3.2 Time Courses and Trajectories. 

Figure 1 exhibits a sublethal (Figure 1A) and a lethal (Figure 1C) time course, and their corresponding trajectories on the *D-S* phase plane (Figure 1B,D, respectively). For this system, *I_X_* = 0.61, so *I* = 0.43 is sublethal and *I* = 0.80 is lethal from (*D_0_*,*S_0_*) = (0,0). *S_max_* is seen to exceed *D_max_* in the sublethal time course, and vice versa in the lethal time course. Each time course begins at (0,0), attains a maximum and then declines over time back to (0,0). For the survival case, returning to (0,0) represents the fading away of both damage and stress responses after injury, and the cell resetting to its preinjury state. For the lethal case, the return to (0,0) represents cell death because the cell disintegrates and all its variables, including *D* and *S*, become zero. The time courses form closed looped trajectories. On the phase plane, trajectories confined to the upper left of the diagonal are always survival trajectories, and trajectories confined to the lower right are always death trajectories, clearly demarcating the phase plane into a survival region and a death region.

### 3.3. Injury Course from (D_0_,S_0_) = (0,0) 

Figure 2A shows *D* and *S* time courses from initial conditions (0,0) across the range 0 < *I* < 2*I_X_*. The form of the time courses varies as a function of *I*. The duration taken by the time courses to return to (0,0) increases as *I* approaches *I_X_* from both the left and the right. Figure 2B plots *D_max_* and *S_max_* vs. *I* from the time courses shown in Figure 1A and provides a summary of the injury dynamics of this system from initial conditions (0,0). The maximum curves cross at *I_X_*. To the left of *I_X_*, *S_max_* > *D_max_*, and to the right of *I_X_*, *D_max_* > *S_max_*. 

Figure 2C extends the *I*-range to 0 < *I* < 5*I_X_* to illustrate that *I_max_* ~ 3, where *I_max_* is defined as the *I* value in which the *S* time course is always zero. *I_max_* is of formal value by setting the upper bound of the *I*-range for a given input vector. The lower bound of *I* is of course zero. In this example, *I_max_* is ~6 times larger than *I_X_*. 

### 3.4. Time Courses over Ranges of Initial Conditions 

The effect of initial conditions on the injury dynamics is an important factor because it can potentially reverse outcome. Figure 3 illustrates the effect of altered initial conditions on the sublethal time course from *I* = 0.43129 (shown in Figure 1). In Figure 3A, the nine time courses correspond to the initial conditions marked by dots in Figure 3B. The green dots in Figure 3B indicate *S_max_* > *D_max_* (the survival outcome) and the red dots indicate *D_max_* > *S_max_* (the death outcome) for the corresponding time courses. Three of the nine initial conditions caused the survival outcome to “flip state” to death outcomes.

The result is sensible. The three time courses that flipped outcome started with *D_0_* positive, which is interpreted as inducing some form of damage in the cells *before* applying injury *I,* i.e., a *pretreatment*. For example, one could imagine applying a mitochondrial inhibitor in sublethal doses before applying a second drug, for example, thapsigargin. The increase in *D_0_* would correspond to increasing sublethal doses of the mitochondrial inhibitor. This will certainly weaken the cells. Then, application of the thapsigargin at a sublethal dose induces the injury dynamics in cells with pre-existing damage. The sum of the pre-existing damage and damage from the thapsigargin kills the cells, even though application of the thapsigargin by itself would not. From this example, we gain insight into how application of multiple treatments to cells is by no means neutral and that application of any agent intimately affects the cell’s injury dynamics. 

In Figure 3C, instead of nine time courses, 2500 time courses (50 *D_0_* by 50 *S_0_*) were calculated and outcome plotted as indicated above, filling in the plane of initial conditions. Calculating a large number of time courses caused survival and death regions to become visible on the initial conditions plane. The region size can be quantified by the ratio of death outcomes to all outcomes. In this example, 47.25% of initial conditions resulted in a death outcome (the remaining 52.75% gave survival outcomes). We emphasize that the injury is *sublethal* from initial conditions (0, 0). However, given a range of meaningful initial conditions, almost *half* of the time courses result in killing the cells under supposedly sublethal conditions.

This example illustrates how theory allows us to explore conditions that present practical obstacles to measure. While measuring 9 time courses is feasible, measuring 2500 would require some type of automated method and could not be performed by hand.

### 3.5 Injury Course over Ranges of Initial Conditions 

We can construct initial condition “outcome planes” at each value of *I* in the injury course. Figure 4A shows such “outcome planes” for one sublethal and two lethal values of *I*. We saw above that a sublethal plane can produce death outcomes under some initial conditions. Similarly, lethal planes can produce survival outcomes at some initial conditions. Again, the result is sensible. When *S_0_* is increased over *D_0_*, the pre-activated stress responses mitigate the injury *I* and cause survival. This is essentially a preconditioning phenomenon expressed by the theory solutions. We did not seek to specifically make a theory of preconditioning. Instead, from our idealization of acute cell injury in terms of *D* and *S*, and the mathematical forms chosen to represent the idealization, preconditioning emerges as a natural consequence of the dynamics.

As before, we can calculate the percentage of death outcomes on each plane and plot this vs. *I* (Figure 4B). The number of death outcomes increases with *I*, as would be expected. Significantly, from a therapeutic point of view, there are substantial areas of the plane with survival outcomes for lethal injuries. A therapy that could access those regions of the plane would be able to convert a lethal outcome to a survival outcome. The theory thus provides a systematic and quantitative way to access therapy. Further, the therapeutic region changes as a function of *I*, which means “one size does not fit all”. The theory can calculate the required therapy for any specific circumstance. In this example, “therapy” refers to the ranges of initial conditions leading to survival outcomes at *I* > *I_X_*.

### 3.6 Percent Death Plots 

We can calculate as many outcome planes as desired. Figure 5A calculates 20 outcome planes over the range 0 < *I* < 2*I_X_* and the corresponding plot of percent death outcomes is shown in Figure 5B. It is noted that within the 2*I_X_* range, the dynamics are, roughly, 50:50 survival to death outcomes at each *I*. Figure 5C extends the *I* range to 0 < *I* < 5*I_X_* and the percent death outcome is shown in Figure 5D. Now we see that as *I* increases, death outcomes predominate. At *I_max_*, which is approximately 5*I_X_*, there are no longer any survival outcomes. Thus, *I_max_* indicates the end of any potential therapy because all outcomes after *I_max_* are death.

Finally, we wish to briefly illustrate how the injury dynamics can vary from system to system. A given system is mainly defined by the 4 qualitative parameters (*c_D_*, *λ_D_*, *c_S_*, *λ_S_*). Varying (*c_S_*, *λ_S_*) indicates a different cell type, and varying (*c_D_*, *λ_D_*) indicates a different injury mechanism. To model a different cell type, *c_S_* is increased from 0.4 to 4, corresponding to a cell with stronger stress responses. The initial condition outcome planes (Figure 5E) and percent death outcome plot (Figure 5F) exhibit different dynamics from the *c_S_* = 0.4 case. Notably, the percent death outcome plots are different between the two systems. For *c_S_* = 0.4, below *I_X_*, the outcomes are roughly 50:50. For the *c_S_* = 4 case, the results are nonmonotonic and unexpected: there is (1) a narrow range below *I_X_* where there is close to 100% survival outcomes, and (2) a range at very low *I* where death outcomes are increased when *I* decreases. Also, for *c_S_* =4, *I_max_* occurs ~ 2*I_X_* vs. 5*I_X_* for the *c_S_* = 0.4 case, meaning it has a relatively more compressed *I*-range.

Comparing two input vectors illustrates two points. First it underscores the need to systematically study how Equation (5) behaves as the 4 qualitative parameters vary. We are currently undertaking this task and it will be the subject of a future report. Second, this comparison illustrates how two different cell types exhibit very different dynamics to injury. What this means in practical terms is that one cannot assume that because cell type A behaves in a certain fashion when injured, cell type B will behave similarly. This is a well-appreciated insight with respect to tissue differences, for example, injured brain versus injured heart. However, it is less acknowledged in cell culture studies. Our results quantitatively demonstrate that cell type variations can widely alter injury dynamics.

## 4. Discussion

We showed here how to solve the nonautonomous theory of cell injury dynamics and interpret solutions of Equation (5) in terms of acute cell injury. The results obtained illustrate that the theory produces output that is both sensible and insightful with respect to known phenomena associated with acute cell injury such as preconditioning, or variations in the length of time it takes a system to die after injury (e.g., necrosis vs DND in stroke). Our nonautonomous theory demonstrates that outcome is a function of injury intensity and that *D* and *S* time courses are, in general, different for different injury intensity *I*. We also demonstrated that, in general, the range of initial conditions resulting in survival outcomes at lethal *I* > *I_X_* decrease as *I* increases, until *I_max_*, after which all outcomes are death. These results clearly specify that a “one size fits all” therapy will be unsuccessful at effecting survival at all lethal injury values. Below, we compare the output of Equation (5) to our previous autonomous version of theory, and then conclude with statements about the value of abstraction and idealization for biomedical sciences.

### 4.1 Outcomes in the Autonomous vs. Nonautonomous Theories

The technical differences between autonomous and nonautonomous differential equations mean there is not a direct one-to-one mapping of the solutions. However, there are features of the solutions that are analogs in terms of how they are interpreted. For example, the autonomous theory output fixed points (*D**, *S**) at each value of *I*, and injury courses were expressed as plots of fixed points vs. *I* [4]. Such plots are called bifurcation diagrams. Technically, there is only one fixed point for all solutions to the nonautonomous theory: (*D**, *S**) = (0, 0) as *t* → ∞. The nonautonomous theory was designed to have this feature, which is a necessary condition to have closed loop trajectories (Figure 1B,D). Therefore, fixed points cannot be directly compared between the two versions of the theory. Instead, the maximum points of the time courses (*D_max_*, *S_max_*) from the nonautonomous version are functionally analogous to the fixed points of the autonomous theory. Plotting maximum points vs. *I* (Figure 2B,C) resemble a bifurcation diagram. However, Figure 2B,C are not bifurcation diagrams because a bifurcation diagram accounts for all initial conditions. The percent death plots (Figure 4B and Figure 5B,D,F) are thus the analogs of bifurcation diagrams for the nonautonomous theory because they incorporate all different initial conditions.

It needs to be stated that percent death plots are **not** to be interpreted that, given a specific value of *I*, there is a probability of X % that the cell will die. There is nothing statistical about the theory. It must be firmly kept in mind that any time course from a specific initial condition is a deterministic outcome. The percent death plots are meant only as summaries of all the time courses from all the initial conditions at a given value of *I*. Given some value of *I* and specific initial conditions, one can then calculate the specific deterministic time course. Do we expect that the precision to identify point-like initial conditions is possible in biomedical studies? Certainly not with today’s technology. However, we do not need point precision. As the “outcome planes” indicate, outcome is associated with ranges of initial conditions, and such ranges are likely good enough to map to current experimental technologies.

### 4.2 Injury Courses in the Nonautonomous Theory 

The functional injury course for the nonautonomous theory is the percent death plot (e.g., Figure 4B and Figure 5B,D,F). A percent death plot allows assessment, at a glance, of survival across the entire range of injury intensities. From the two examples calculated above (Figure 5D,F) we conclude that, in general, different percent death plots are obtained from different input vectors. Knowledge of how the percent death plots vary with (*c_D_*, *λ_D_*, *c_S_*, *λ_S_*) will be important to fully systematize the nonautonomous theory. For the autonomous version, only four qualitative types of bifurcation diagrams were observed [4]. Thus, in the scope of the autonomous model, there were only four basic forms of acute injury dynamics. It remains to be seen if a similar simplification of injury dynamics occurs in the nonautonomous theory and therefore a parameter sweep study is important to undertake.

However, even without a complete understanding of the dynamics of Equation (5), we can still make important and relevant comments about the meaning of the percent death plots and how they compare to the injury courses of the autonomous theory. Figure 6 shows two of the four types of bifurcation diagrams obtained from the autonomous version (the other two types are variants of Figure 6B and are not discussed here). Figure 6A is monostable, meaning only a single pair of fixed points (*D**, *S**) at each *I*. The interpretation of this type of injury course is that for all *I* < *I_X_*, the cell always survives, and for all *I* > *I_X_*, the cell always dies. There is no therapy possible in this type of injury dynamics. It is unrealistic because a sublethal injury damaged by a pretreatment (e.g., *D_0_* > 0) will not kill the cell, no matter how strong the pretreatment damage (e.g., even *D_0_* → ∞ would lead to survival of the cell). Similarly, no pre-inhibition of damage (*D_0_* < 0) or pre-activation of stress responses (*S_0_* > 0) will halt cell death when *I* > *I_X_*. The monostable case is completely ideal and does not correspond to reality. It does capture, however, the common idea of a “cell death threshold” where the cell survives below the “cell death threshold” and dies above the “cell death threshold”.

On the other hand, the bistable injury course does allow for death at *I* < *I_X_*, and survival for *I* > *I_X_* in the bistable regions where both solutions simultaneously exist in the system dynamics (Figure 6B, where area marked in yellow is the bistable region). This was, in fact, the central insight of the autonomous model: that *therapy was only possible when the dynamics were bistable*. This was a major finding with respect to the link between injury dynamics and therapeutics. We have stated elsewhere and repeat here that this is perhaps the most important insight provided by our theoretical study because *it opens the possibility to calculating therapy for any given situation*.

What is of great interest, and is the main finding of the present study, is the following: With respect to the nonautonomous theory, *the system is, in general, “bistable” at all values of I* < *I_max_*. Again, because of the technical mathematical differences, the term bistable is inappropriate and used only by analogy in the following sense. What is demonstrated in Figure 5D,F is that, at each value of *I* < *I_max_*, there exist time courses across the initial conditions with both survival and death outcomes. At a given *I*, the “outcome plane” was clearly demarcated into a survival region and a death region, and access to each region is granted by application of the appropriate initial conditions. Further, the area of the death region on the outcome planes increased with *I*, until it subsumed 100% of the plane at *I*
>
*I_max_*. This result provides a considerably more realistic model of cell injury dynamics.

### 4.3 Sublethal and Lethal Conditions Form A Continuum 

Above we stated that specific qualitative responses to cell injury, such as rapid or delayed preconditioning, or necrosis or delayed neuronal death, are cross sections of a continuum of injury dynamics. This point is made succinctly in Figure 3A showing different time courses at the *same value of I*. The time courses are distinguished by their initial conditions, which again, correspond to the variety of manipulations performed on cells or tissues in the laboratory. It is clearly seen that the time courses can have different forms and durations. This kind of complexity is not intuitive to the current biomedical paradigms.

The continuum of responses is also illustrated in Figure 2A that shows a series of time courses across the *I*-range starting from (*D_0_*, *S_0_*) = (0, 0). With respect to sublethal effects, the area under the curves of the *S* time courses (i.e., the accumulation of *S* over time) is greater than the areas under the *D* time course. This indicates there is excess stress response beyond that needed to inhibit the actual damage. This excess stress response is what causes the preconditioning response. If a second injury is applied at some later time before the excess stress response fades, the excess stress response is available to combat the second injury and ameliorate its effect, allowing the cell to survive. Further, the area under the *S* time courses decreases from *I* = 0 to *I* = *I_X_*. This indicates that preconditioning will be a graded response and a function of *I*. This is well-known in the empirical literature. There is an optimal sublethal injury required to produce the optimal preconditioning effect. 

Now consider the lethal side of the time courses in Figure 2A. For those time courses close to *I_X_*, it takes a much longer time for the system to decay to death, which would be a DND phenotype in the case of stroke. At the highest *I* values, the time course returns to (0,0) very rapidly, and this would correspond to necrotic forms of cell death. Thus, the theory reveals that these are not different forms of cell death, but cross sections along the continuum of injury dynamics that are a function of *I*. 

Therefore, a bona fide mathematical theory of cell injury dynamics demonstrates that the variations in survival responses (e.g., preconditioning), and death phenotypes are intimately interlinked and form a continuum of states.

## 5. Measuring the Theory

The main purposes of this paper have been: (1) to show how to solve the nonautonomous theory and interpret the solutions in terms of acute cell injury, and (2) illustrate how to think of injured cells in abstract, ideal terms. We have not focused on how we would measure or test the theory. In the following we make a few comments along these lines.

The link between the theory and real injured cells or tissues are the concepts of *D* and *S*. When a brain or heart is injured by ischemia, or when cultured cells are injured by thapsigargin (or any other injury mechanism), one must envision that *D* and *S* are real phenomena occurring inside the cells and that the amount of each follows time courses as calculated by the theory. Ideally, one would then measure how *D* and *S* change with time and determine if the theory accurately predicts the *D* and *S* time courses. 

The empirical question thus comes down to measuring *D* and *S*. Recalling their definitions, *D* is the total damage and *S* is the total-induced stress responses. Thus, to measure them would be to measure every single form of post-injury damage at a point in time and sum them together to obtain *D* at that time point. Similarly, every induced stress response would be measured and summed to obtain *S*. In practice, given today’s technology and our incomplete understanding of cell physiology, this is impossible.

However, taking this approach is analogous to measuring temperature by attempting to measure the velocity of every single particle in the medium and average them, which is also impossible. Instead, we measure temperature via e.g., the expansion volume of a liquid, typically mercury. This provides a surrogate measure that correlates perfectly with the average velocity of the particles making up the medium whose temperature we seek to know.

A similar approach is required to estimate *D* and *S*. There must be specific changes inside the injured cells that estimate or track the real values of *D* and *S*. We have hypothesized that the gene changes inside the cell track *S* and that some general form of cell damage, such as protein aggregates, track the total damage *D*. We have made these measurements and are in the process of preparing our results for publication. We have not included our empirical work here because it is outside the scope of the present work, and the present work is a necessary prelude for reporting our empirical results.

The important point to emphasize here is that there is a chasm between the mainstream descriptive approach to cell injury that assumes it can discover some qualitative feature that causes cell death, and our theoretical approach that dictates *a priori* what needs to be measured to characterize cell injury. In our theory, *D* and *S* are *a priori* concepts, and the theory indicates that empirical work needs to be directed first to discover how to quantitatively estimate *D* and *S*, and then determine their time courses. Ultimately, it is the failure of the descriptive approach that has motivated us, and we believe, in the long run, both approaches will be necessary much in the same manner that theory and experiment co-exist in physics.

## 6. Conclusions

We have shown here a plausible and effective way to express solutions from the nonautonomous theory of acute cell injury dynamics. The solutions are sensible and capture important aspects of real behaviors observed in acutely injured biological systems. The solutions to the nonautonomous version are more realistic than the autonomous version, as discussed above. Both versions of the model possess important implications for therapy and indicate the possibility to calculate therapies for specific acute injuries with engineering-like precision.

Our goals here were two-fold: first to show how to solve and interpret the nonautonomous version of the dynamical theory of acute cell injury and second, to use this to illustrate the value of abstraction and idealization for biomedical science. In a sense, it is immaterial whether Equation (5) is correct or not. We have been explicit in our assumptions of the mathematical forms used and these can be modified as necessary to better fit real data from real injured systems. In an upcoming work we will discuss our attempt to experimentally measure *D* and *S* time courses and fit them to the nonautonomous cell injury theory. Any weakness in the specific mathematical formulation is offset by the framework and the potential it provides to organize and systematize the study of cell injury across all biomedical fields that involve acute injury.

## Figures and Tables

**Figure 1 brainsci-08-00039-f001:**
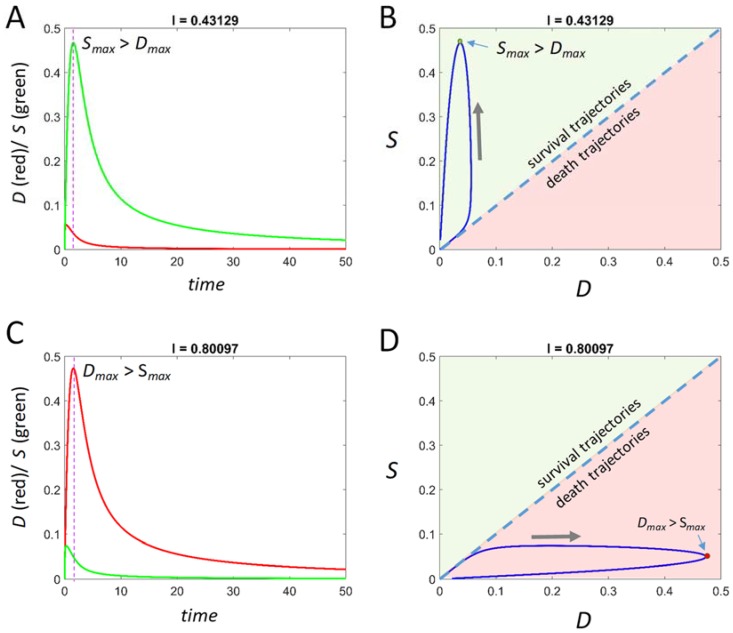
Example time courses calculated from Equation (5). (**A**) A pro-survival time course and (**B**) its corresponding trajectory on the phase plane. (**C**) A pro-death time course and (**D**) its corresponding trajectory on the phase plane. Purple dashed lines mark maximum values of time course in (**A**,**C**), and these are marked by green and red dots in (**B**,**D**), respectively. Blue dashed lines are the diagonal of the phase plane. Gray arrows indicate the flow direction of the trajectories. For this system, *I_X_* = 0.61.

**Figure 2 brainsci-08-00039-f002:**
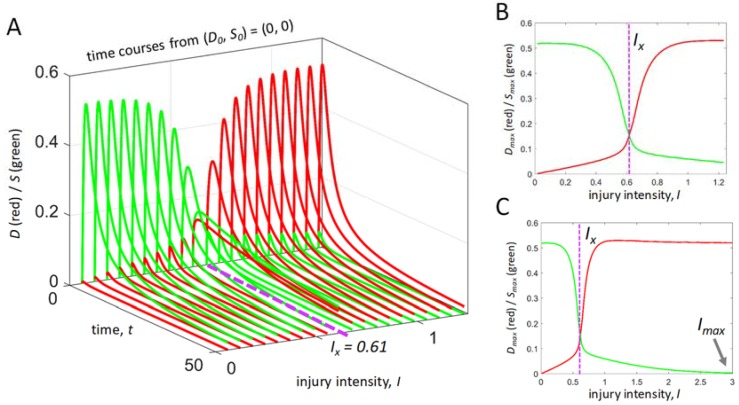
The injury course from initial conditions (0,0). (**A**) *D* and *S* time courses across the range 0 < *I* < 2*I_X_*. (**B**) Plots of *S_max_* (green) and *D_max_* (red) vs. *I*, for the time courses shown in A. (**C**) Plots of *S_max_* and *D_max_* vs. *I* across the range 0 < *I* < 5*I_X_* illustrating *I_max_*. Purple dashed lines indicate the position of *I_X_* on the injury course.

**Figure 3 brainsci-08-00039-f003:**
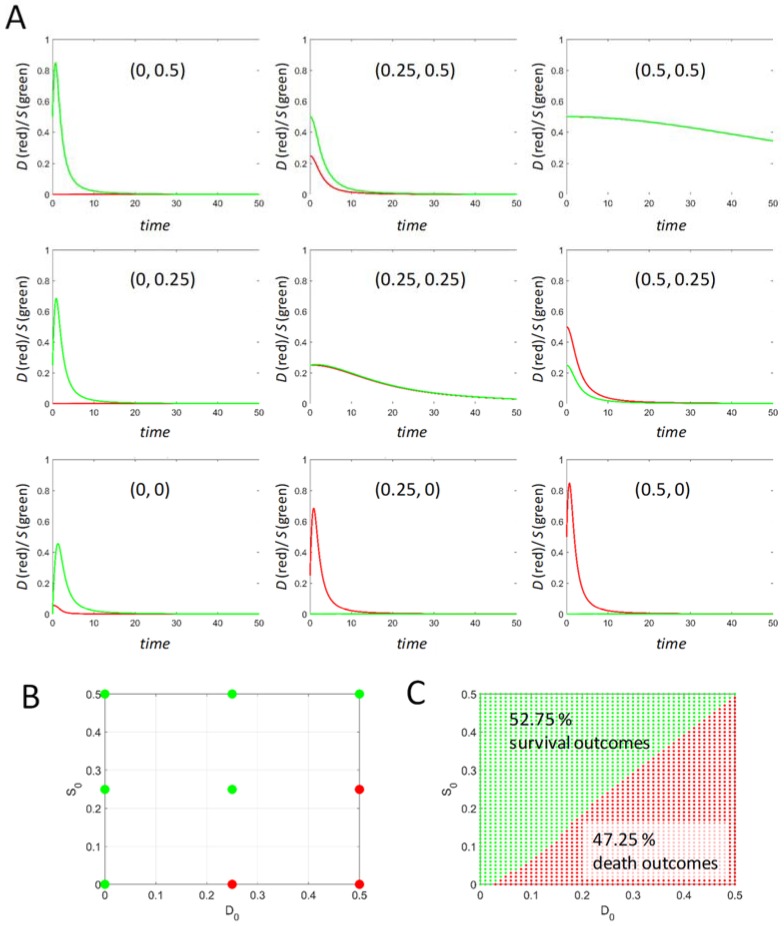
: Time courses at *I* = 0.43 < *I_X_*, from varying initial conditions. (**A**) A series of nine *D* and *S* time courses started from initial conditions (*D_0_*, *S_0_*) as listed on each plot. (**B**) The plane of initial conditions used for the time courses in A, where the outcomes are indicated by green (survival) or red (death) dots. (**C**) The same plane in B filled in with 2500 time course outcomes demarcating a survival and a death region with respect to initial conditions.

**Figure 4 brainsci-08-00039-f004:**
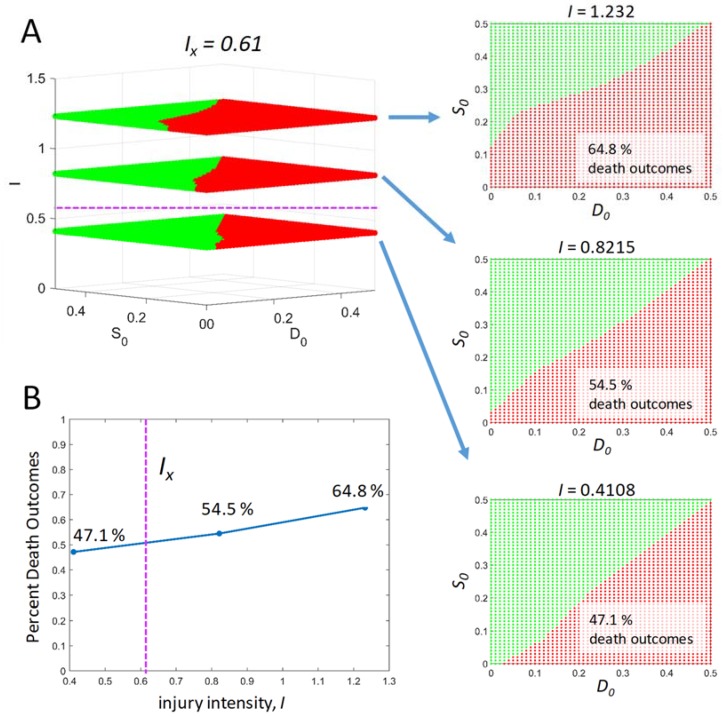
Initial condition outcome planes as a function of *I*. (**A**) Three outcome planes plotted vs. *I*. Arrows point to face-on views of each plane. (**B**) Percent death outcome for each of the three planes vs. *I*.

**Figure 5 brainsci-08-00039-f005:**
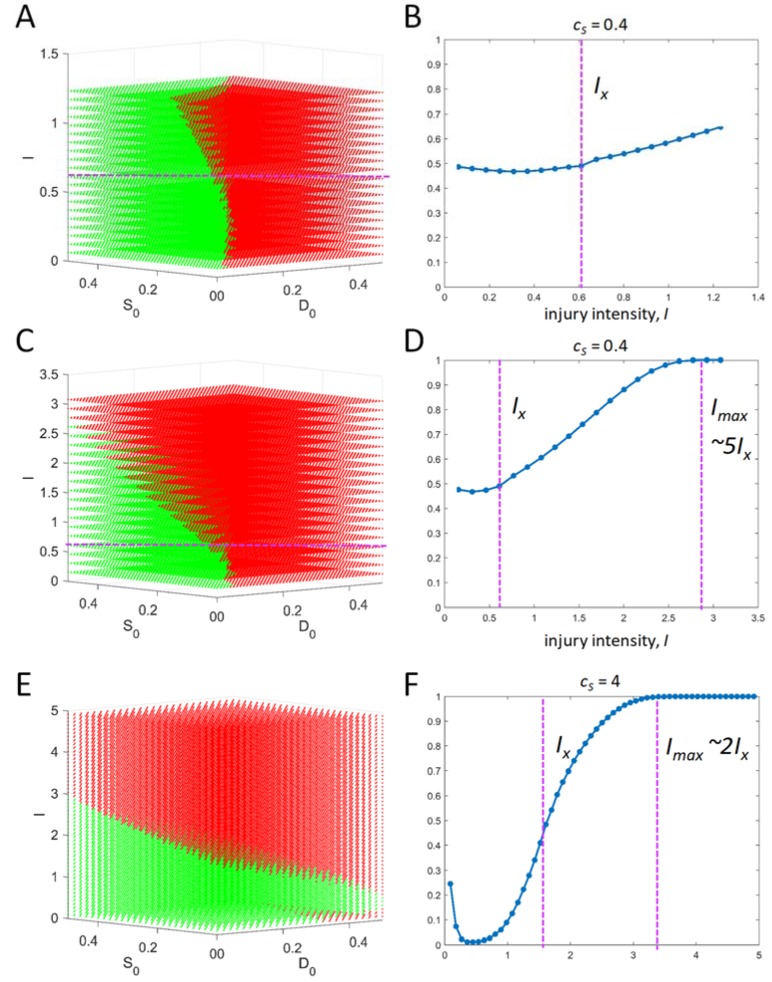
(**A**) 20 outcome planes plotted from 0 < *I* < 2*I_X_* for input vector with *c_S_* = 0.4. (**C**) 20 outcome planes plotted from 0 < *I* < 5*I_X_* for the input vector used in (**A**). (**E**) Outcome planes for an input vector with *c_S_* = 4. The corresponding percent death plots are shown in (**B**), (**D**), and (**F**) respectively. Purple dashed lines in (**B**), (**D**), and (**F**) indicate *I_X_* (left) and *I_max_* (right).

**Figure 6 brainsci-08-00039-f006:**
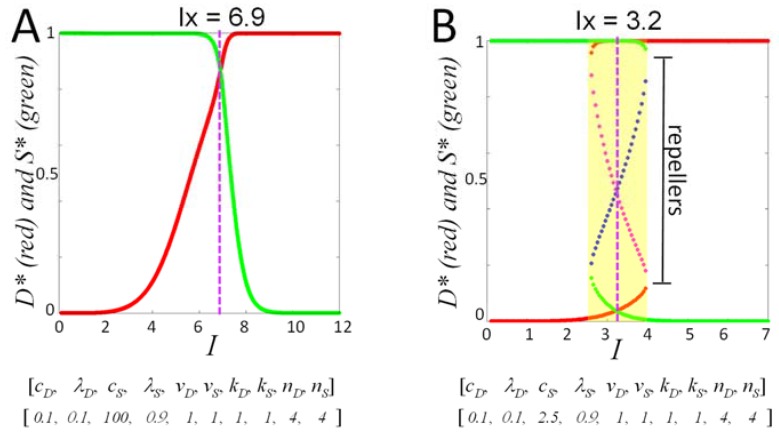
(**A**) Monostable and (**B**) bistable bifurcation diagrams from the autonomous dynamical theory of acute cell injury. Purple lines mark *Ix*. Input vectors are shown below the respective injury courses.

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
