# Peer review of "Abstraction and Idealization in Biomedicine: The Nonautonomous Theory of Acute Cell Injury"

_brainsci, 2018, doi:10.3390/brainsci8030039_

Round 1

Reviewer 1 Report

The manuscript is interesting and I will recommend the acceptance after a following minor revision.

1. Some letters in each figure are too small.

Author Response

Reviewer 2 requested we increase the font size on the figures. We have gone through Figures 1 through 5 and increased fonts where the text on the figures conveyed important information. We did not change Figure 6 since there was not an apparent problem with the font sizes in that figure.

Reviewer 2 Report

The study is a continuation of theoretical modeling of cell death by this research group. The paper extended their previous calculations by applying autonomous and non-autonomous nonlinear theories in understanding outcomes of cell death or  survival in response to acute insult. Overall, the paper was well written, and  contained detailed descriptions of the calculation and graphs. There are several concerns:

The ultimate goal of the study would be to develop the theories which are testable by experiments. However, the paper presented limited discussions to link to experimental biology. In the introduction, some concepts were included, such as type of cell death, sub lethal insults (preconditioning), etc, but, at the end of the paper, no discussions were included to relay to these issues.  In addition, in the middle of modeling in the paper, mitochondrial inhibition  and ER stress inducer were causally mentioned, which is not rigorous because both factors could cause cell death with increased exposure time.

Taken together, I felt that the study should be constructed in a fashion which will benefit for experimental biology readers.  The current version has many limitations to reach a broad readership.

Author Response

Reviewer 1 asked us: (1) to clarify the discussion linking initial conditions and the experimental example (e.g., in section 2.3 “Initial Conditions”), and (2) to discuss aspects “which will benefit for experimental biology readers”.

In reply to Reviewer 1, we note that one of our two main intentions in this paper was to expose experimental biologists to the abstractions and idealizations that are necessary for theoretical work. Therefore, focusing only to experimental issues would negate this expressed purpose of the manuscript. Section 2.3 was originally included to illustrate an important example where our theory meets typical biomedical experimental designs. Many types of manipulations are performed on cells and tissues that amount to changing the initial conditions. We illustrate (particularly in Figure 3) that such manipulations can wholesale change the system dynamics. In this revision, we made the discussion in section 2.3 clearer to show the links between the theory and typical experimental practices, and added some further explanation and summary in the discussion section. We hope these address Reviewer 1’s concerns.

To address experimental biology readers, we also added a new section 5 “Measuring the Theory” that briefly discusses the direct link between the theory and its empirical confirmation. We took this as an opportunity to expand on our intention to discuss abstraction and idealization in biomedical studies. We explain how it is the theory that dictates what is to be measured, which in this case are D and S, and briefly outline important theoretical and empirical aspects of measuring these.

As explained in the new section 5, we are preparing our empirical estimates of D and S, but to go into too much detail in the present manuscript would overly complicate as well as dilute our demonstration of how to solve and interpret the nonautonomous theory, which, as we stated there, is a necessary prerequisite to the empirical work.